# The Current Progress and Future Options of Multiple Therapy and Potential Biomarkers for Muscle-Invasive Bladder Cancer

**DOI:** 10.3390/biomedicines11020539

**Published:** 2023-02-13

**Authors:** Ying Shi, Bryan J. Mathis, Yayun He, Xiong Yang

**Affiliations:** 1Department of Urology, Wuhan Union Hospital, Huazhong University of Science and Technology, Wuhan 430022, China; 2International Medical Center, University of Tsukuba Hospital, Tsukuba 305-8576, Ibaraki, Japan; 3Department of Urology, The Second Hospital of Wuhan Iron and Steel (Group) Corporation, Wuhan 430082, China

**Keywords:** muscle-invasive bladder cancer, diagnosis, therapeutics, exosome, biomarker

## Abstract

Bladder cancer is a common disease in men and the elderly. Current treatment paradigms include radical resection of the bladder and lymph nodes or transurethral resection, both supported by chemotherapy and/or radiation. New modalities, such as illumination-based therapies are also being translationally pursued. However, while survival rates have increased due to combined therapies (particularly chemotherapy, radiation, immune checkpoint inhibitors, and surgery), a lack of diagnostic markers leads clinical professionals to rely on frequently invasive and expensive means of monitoring, such as magnetic resonance imaging or bladder cystoscopy. To improve real-time diagnostic capabilities, biomarkers that reflect both the metabolic and metastatic potential of tumor cells are needed. Furthermore, indicators of therapy resistance would allow for rapid changes in treatment to optimize survival outcomes. Fortunately, the presence of nanoscale extracellular vesicles in the blood, urine, and other peripheral fluids allow for proteomic, genomic, and transcriptomic analyses while limiting the invasiveness of frequent sampling. This review provides an overview of the pathogenesis and progression of bladder cancer, standard treatments and outcomes, some novel treatment studies, and the current status of biomarker and therapy development featuring exosome-based analysis and engineering.

## 1. Introduction

Bladder cancer remains a serious disease with only incremental progress in curative therapies. Men remain far more susceptible to bladder cancer over their lifetimes (hazard ratio 1.08 for men, 0.27 for women) and such malignancies are the 6th most common cancer in European older men (median age 70) [1]. In new bladder cancer patients, roughly 75% start with non-muscle invasive bladder cancer (NMIBC) and 10–50% of these patients will progress to muscle-invasive bladder cancer (MIBC) [1]. As the cancer increases in size, it penetrates the lumen, lamia propria, detrusor muscle, and muscularis propria before invading fatty tissue and spreading to the surrounding organs (Figure 1) [2]. Of note, de novo MIBC occurs in 25% of cases and 90% of these will be urothelial carcinomas (UC) while the remainder consists of squamous cell carcinomas or adenocarcinomas [1]. With over 17,000 deaths and 80,000 cases per year (in the US alone), recurrence rates as high as 30–40%, and the threat of metastasis to the thoracic organs, both progressive or de novo MIBC represent a serious threat to clinical outcomes [1,3]. Incremental gains made over the last 20 years in overall survival (OS) and disease-free survival (DFS) rates of MIBC are the result of coupling radical cystectomy and pelvic lymph node dissection (RC/PLND) surgeries with combinations of chemical, immunological, and radiation therapies (multimodal therapy). Predictive scoring for progression and recurrence do exist; however, the inherently unstable nature of MIBC means that treatment is reactive since reliable biomarkers for tumor progression have not been clinically verified. Thus, clinical research is currently evaluating diverse trimodal therapies to improve quality of life in selected patients even as translational research continues to investigate molecular and other biomarkers for use in predicting MIBC progression and potential treatment resistance. 

## 2. The Genetic Basis of Muscle-Invasive Bladder Cancer Pathogenesis

NMIBC progresses to MIBC in a subset of cases due to rapid growth and a putative failure of immune surveillance. In addition to the aryl hydrocarbon receptor in NMIBC development due to chemical exposure, carcinogenic chemicals (such as benzene, nitrosamines [fermented foods or fertilizer runoff], or tobacco smoke) may increase the risk of developing bladder cancer through DNA damage to key checkpoint and oncogenes [1,4]. 

Genetic analyses of MIBC patients point to mutations in FGFR3, PIK3CA, TERT, and p53 for the pathogenesis of NMIBC as well as MIBC progression [3,5]. Additionally, alterations in tumor type are dependent upon these factors, as MIBC seems to be primarily regulated by FGFR3 while NMIBC is more p53 dependent [6]. Heritable factors also play an important role as other genetic studies have found mismatch repair faults (hereditary Lynch syndrome) in MSH2 (and associated genes) to carry higher risk [7]. This was reported in a Swedish study that found MLH1 gene mutations to be present in 40% of affected families, with MSH2 at 36%, MSH6 at 18% and PMS2 at 6% of total cases [7,8]. Lynch syndrome may thus have some influence on bladder cancer pathogenesis due to these oncogene mutations and potential microsatellite instability [8]. However, a study of 164 urothelial carcinoma patients in Japan found a Lynch syndrome prevalence of only 1.8%, similar to a 2017 report in which 6 patients with documented Lynch syndrome were discovered out of 444 total patients (1.3%) while Lynch syndrome-associated neoplasm patients numbered 30/444 (6.7%) [9,10]. Taken together, reports indicate that at least some minor fraction within large populations of UC patients are expected to carry Lynch syndrome (or associated) neoplasms and should be checked for MLH1, MSH2, MSH6, and PMS2 mutations [11].

## 3. Existing Diagnostic and Imaging Paradigms

With regard to bladder cancer screening and treatment evaluation, imaging has stepped in to fill the diagnostic gap made by a lack of reliable biomarkers. These technologies, while continuously refined for improvements in diagnostic accuracy, remain invasive (cystoscopy) or expensive/troublesome for the patient (magnetic resonance imaging and computed tomography).

### 3.1. Imaging Paradigms

#### 3.1.1. Magnetic Resonance Imaging and Computed Tomography

Early and accurate detection of NMIBC, progressive MIBC, or de novo MIBC are crucial for treatment planning. Therefore, magnetic resonance imaging (MRI) or computed tomography (CT) have become gold standards for non-invasive imaging and refinement of various imaging parameters and dyes, especially with regard to MRI, are well-reported. As an example, diffusion-weighted MRI, based on the differences in water molecule Gaussian motion, requires careful tuning of B values to enhance discrimination between tumor and normal tissue [12]. Some reports indicate that diffusion kurtosis, which more carefully analyzes water motion in non-Gaussian systems, plus tumor contact length measurements are more accurate than diffusion weighting [13]. Recently, positron Emission/CT (PET/CT), that relies on higher 2F-18-fluoro-2-deoxy-d-glucose (FDG) uptake by more metabolically active tumor cells, has been reported useful for imaging [14]. While frequent urinary flushing can complicate signal retention within the bladder, FDG-CT is useful for metastatic surveillance of the lymph nodes. Newer dye media, such as C-11 acetate and C-11 choline, exploit the retention of radiotracers that cannot be easily excreted via urine and may also facilitate whole-body scanning for metastatic evaluation during initial screening [14]. 

Even with MRI/CT, diagnostic accuracies from 55–89% are commonly reported and constant evolution, such the development of multiparametric MRI (mpMRI) that does not rely on extensive radiological reading experience, is expected to increase the utility of diagnostic MRI [15]. Unfortunately, the expense and time required to complete imaging sessions is difficult for advanced-stage patients while the data, albeit detailed, is not real-time and is only a snapshot of the physical tumor characteristics. Since prognosis depends on the molecular and metabolic status of the tumor, MRI and PET/CT provides only part of the data needed to plan a complete treatment schedule. 

#### 3.1.2. Cystoscopy

A mainstay of initial diagnosis and treatment, cystoscopy is employed in transurethral resection of bladder tumor (TURBT) procedures for suspected NMIBC as the primary imaging pathway to guide the surgery and for biopsy during follow-up visits. An invasive procedure, it also carries the side effects of pain, potential perforation, and distension of the bladder [16]. In MIBC, cystoscopy is useful only for diagnosis but, unfortunately, cystoscopy alone is only 71% accurate [15]. However, a recent trial (BladderPath, ISRCTN 35296862) is currently evaluating mpMRI for replacement of TURBT staging [16] and neutrophil percentage-to-albumin ratio has also been explored for prognostic potential in MIBC patients treated with neoadjuvant chemotherapy (NAC) and RC [17]. Advances in cystoscopy with photosensitizing agents, such as hexyl aminolaevulinic acid, allow for fluorescent highlighting of tumors to improve detection and ensure precision [18].

### 3.2. Pathology

Histological and histopathological analyses of biopsy specimens remain consistently useful for confirming bladder cancer diagnoses. Paraffin-embedded slicing, fixing, and staining with antibodies specific to driver genes (such as p53 and FGFR3) allow for both aggregate cell counts and morphological confirmation of transformed cells [19]. Morphological and antibody-based classification may be particularly useful in detailing the transformation of NMIBC to MIBC (urothelial to sarcomatoid), especially since PD-L1 expression tends to be higher in MIBC-transformed sarcomatoid cells [20]. However, as with other imaging methods, even the best pathology is a snapshot of a past condition and cannot provide real-time status updates of the molecular and metabolic conditions within a tumor.

## 4. Treatment Paradigms

### 4.1. BCG as the First Line

NMIBC is often first treated with Bacillus Calmette-Guerin (BCG), a biotic therapy originally used as a killed tuberculosis vaccine strain, to train the immune system and evasion of this therapy may result in tumor progression [21]. A first-line adjuvant therapy, BCG treatment is effective in about 50% of patients, with the rest either failing to respond, relapsing, or having adverse events [22]. Originally thought to be wholly immune-centered around CD8^+^ T cells, evidence exists that also demonstrates some direct tumoricidal activity of the bacteria (putatively via oxidative stress or necrotic pathway activation) but increases in PD-L1 expression on tumor cell surfaces might explain BCG evasion [23]. However, for non-responders, valrubicin or pembrolizumab (anti-PD-L1) are the only FDA/European Medicines Agency-approved therapies for NMIBC if BCG is ineffective [24]. 

### 4.2. Surgery Types: Radical Cystectomy plus Neo-Adjuvant Therapy

Although radical cystectomy with pelvic lymph node dissection (RC/PLND) is the current gold standard for MIBC, can result in an R0 (complete cure) condition, and has additional benefits in preventing metastasis, it has side effects that can be devastating for quality of life (incontinence, impotence, neurologic damage) and some patients are unsuitable for such surgery. It is the main line of treatment for progressive or de novo MIBC; however, RC alone is also not a definitive treatment, as 5-year survival rates of 40–60% and recurrence in as little as 12 months have been reported [25]. For this reason, RC is usually combined with neo-adjuvant chemotherapy (NAC) to maximize tumor control before and after surgery. Neo-adjuvant cisplatin with RC is the currently recommended multimodal treatment standard for MIBC [26]. 

While the use of chemotherapy with RC seems obvious, NAC has been reported as underutilized, with less than 20% of patients in the US between 2004–2014 having received it [27]. Meanwhile, a recent meta-study of 35,738 patients in 13 reports found that only 17.2% of patients underwent NAC regimens [28]. Since complete, partial, and downstaged response rates were 16.6%, 14.6%, and 45.0% in that study, NAC may be an important weapon against UC progression [28]. Another meta-study of 8 reports found NAC + RC was superior to RC alone with regard to overall survival (OS; HR 0.79; 95% CI: 0.68–0.92, *p* = 0.002), bolstering the utility of NAC to provide improved MIBC prognoses [29]. 

### 4.3. Trimodal Bladder-Preserving Treatment (TMT)

Trimodal therapy (TMT), consisting of combined chemo- and radiotherapies plus TURBT, has been adopted as an alternative to RC/PLND in selected patients [30]. Such bladder-sparing improves quality of life but requires optimal patient selection with regard to co-morbidities and tumor status (ideally cT2 with no carcinoma in situ) [31]. However, an insufficient number of clinical trials prevents the full clarification of survival and quality-of-life improvements that may be possible with TMT. A recent report simulated 500,000 patients with a 2-D Markov model and, in comparisons between RC and TMT, found that TMT had slightly higher quality of life in elderly patients while RC had better overall survival and life quality in younger patients [30]. Similarly, a US study of 2306 military veterans with MIBC found that TMT was associated with comparable survival to RC with neoadjuvant chemotherapy but only in patients older than 65 years of age [32]. Thus, the utility of TMT may be comparable at best in some patients but worse in older patients. Additionally, a 2018 meta-study of 57 total studies and 30,293 patients found that, while TMT mean 10-year OS was insignificantly lower than RC (30.9% TMT vs. 35.1% RC, *p* = 0.32), it was chemotherapy response that determined the best survival results with TMT [33]. The concept of RC for long-term survival superiority was also questioned by a 2020 metasudy by Ding et al that found superior OS results for RC after 10 years but only when Charlson comorbidity scores were 0 [34]. Furthermore, within that 10-year timeframe, TMT was comparable, indicating that TMT is a valid therapeutic option for patients who are unsuited for RC or who do not wish to undergo radical resection [34]. 

These data indicate that, while R0 resection is considered curative, long-term survival also depends on a complete response to chemotherapy and selection of chemoagents by using predictive models for response is therefore a critical component of improving TMT performance. Several proposed and current trials that may finely tune response rate predictive models are currently exploring combinations of immune checkpoint inhibitors and radiation for cisplatin and RC-ineligible patients [35,36]. As such, maintaining a maximum level of tumor control with TMT requires accurate and precise biomarkers to select the proper chemotherapy agents as well as to monitor progress after treatment. The current requirement of frequent CT/MRI or cystoscopy to monitor progress is not clinically feasible and cannot accurately predict resistance to therapy. Outcomes from clinical studies for TMT published after 2010 are summarized in Table 1.

### 4.4. Immune Checkpoint Inhibitors

Since atezolizumab was first approved by FDA for metastatic UC in 2016, immune checkpoint inhibitors (ICIs) to PD-1, PD-L1, or CTLA-4, which target angiogenesis and sensitize the immune response to limit both tumor growth and metastasis, are frequently employed as an adjuvant therapy combined with chemotherapy. Anti-PD-L1 therapy on cancer cells prevents binding of PD-1 on T cells to increase apoptosis while CTLA-4 blocks CD80 and CD86 activity, downregulating Treg-mediated immunoregulation and increasing CD8^+^ cytotoxic activity [44]. In this fashion, antitumor T cell activity is maximized. Both types of drugs are often combined (ICI-ICI) to starve the tumor and increase immune effectiveness since a metastudy of 2 RCTs with 1518 total patients found ICI treatment alone (atezolizumab or nivolumab) was not significantly useful in high-risk muscle-invasive UC [45]. Conversely, a study (CheckMate 032) of combined PD-1 and CTLA-4 inhibitors (nivolumab 1 mg/kg plus ipilimumab 3 mg/kg) found response rate of 38.0% in 92 patients receiving both drugs [46]. Adverse events must also be considered, as a metastudy of 21 reports with 11,454 patients total who received nivolumab, pembrolizumab, atezolizumab, or ipilimumab found a higher incidence of non-fatal adverse events [47]. With the promising results of adjuvant immunotherapy, clinical trials of neoadjunctive immunotherapy have been intensively carried out, and Table 2 summarizes such trials. 

Much effort has been made to look for reliable biomarkers for predicting response to ICIs, such as PD-L1 or CD8 expression by immunohistochemistry [48,49,50,51] and/or tumor mutation burden (TMB) [52]. Unfortunately, these biomarkers have not yet to be verified in large clinical trials [51,53]. 

**Table 2 biomedicines-11-00539-t002:** Outcomes from Select Clinical Studies of neoadjuvant immunotherapy for MIBC.

Trial	Phase	Regimen	Patients	N	pCR%
PURE-01 [54]	II	Pembrolizumab	cT ≤ 3bN0	114	37
ABACUS [55]	II	Atezolizumab	cT2-4N0M0	88	31
NABUCCO *	Ib	Nivolumab + ipilimumab	cT3-T4aN0M0 /T1-4aN1M0	54	Ipi-high 63 Ipi-low 29
BLASST-1 **	II	Nivolumab + GC	cT2-4aN ≤ 1M0	43	49
GU14-188 Cohort 1 ***	II	Pembrolizumab + GC	cT2-4N0M0	43	44.4
SAKK 06/17 #	II	Durvalumab + GC	cT2-4aN ≤ 1M0	53	34
RACE IT ##	II	Nivolumab + Radiotherapy	cT3-4N ≤ 1M0	33	38.7

* 2022 ESMO Poster session 18 Abstract 1770P. ** 2020 ASCO-GU. Abstract 439. *** 2020 ASCO. Abstract 5047. # 2022 ASCO. Abstract 4515. ## Annals of Oncology (2022) 33 (suppl_7): S808–S869. GC = gemcitabine + cisplatin chemotherapy.

### 4.5. FGFR Inhibitors

As mentioned in Section 2, FGFR3 is a common mutation found in patients with MIBC [56]. Erdafitinib, a pan-FGFR tyrosine kinase inhibitor, is the first FDA-approved targeted therapy for mUC with susceptible FGFR2/3 alterations following platinum-containing chemotherapy. A phase II trial of 99 enrolled patients with local advanced and unresectable/metastatic UC with an FGFR3 mutation or FGFR2/3 fusion observed disease progression in all patients following chemotherapy [57]. The confirmed response rate was 40% while an additional 39% of patients were stabilized. For 22 patients with previous immunotherapy, the erdafitinib response rate was 59%. At 24 months median follow-up, the median overall survival was 11.3 months. Adverse events of >grade 3 related to treatment occurred in 46% of patients while 13% had to discontinue erdafitinib due to adverse events [57]. Based on those results, several other FGFR inhibitors are being evaluated, including infigratinib, which has demonstrated promising activity [58]. 

### 4.6. Future Treatments Compatible with TMT and RC

Even with an absence of viable biomarkers for prediction and treatment management, development of new modalities for bladder cancer continue to increase specificity by combining drugs with antibody conjugates (antibody-conjugated drugs; ADC). Unlike ICIs, these drugs are manufactured to deliver cytotoxic molecules via antibodies (usually IgG) engineered for strong interaction with specific tumor antigens and very little cross-reactivity [59]. Connecting links between the antibody and payload are usually constructed of disulfide- or protease-dependent bonds in order to prevent premature release of the payload and to exploit the acidic pH and ROS-intensive microenvironments of tumors that are likely to sever the linker [59]. Then, cytotoxic molecules such as auristatins (microtubule destabilizers), maytansinoids, DNA alkylators, or proteotoxins (protein synthesis inhibitors) are connected to the scaffold as effector payloads to slow tumor cell growth and prevent replication. Other antibody-based fusion techniques concurrently being developed to stimulate immunogenic responses (e.g., ALT-803) and deliver current ICIs with higher specificity (e.g., ATOR-1015) have been extensively reviewed by Bogen et al. [60].

With the discovery of fucolsylated glycans as potential biomarkers for pancreatic, MIBC, and other cancers, the potential of lectin-targeted payload delivery to MIBC has become feasible [61,62]. Lectin specificity to these glycans is possible with recombinant technology and future sequencing studies on diverse bladder cancer specimens could allow for a glycan-lectin binding library to be constructed for targeting both non-invasive and invasive bladder cancers [63]. Additionally, these lectins are amenable to conjugation with nontoxic, photoreactive dyes that respond to near-infrared (NIR) light by conformational changes that induce necrotic death in cells [64]. A recent paper by Kuroda et al. has demonstrated the feasibility of this system for pancreatic cancer in a murine model [65]. Since NIR is harmless to human tissue and cystoscopes already have illumination capability, addition of NIR light and lectin-conjugated dye systems to TURBT may be a possibility that removes residual disease and promotes complete response.

## 5. Scoring Problems and the Search for Biomarkers

A scoring system from the European Organization for Research and Treatment of Cancer (EORTC) has attempted to factor in stage, CIS, tumor grade, size, multifocality, and prior recurrence to create a predictive instrument for progression and recurrence of NMIBC [66]. Other scoring systems take into account responsiveness to BCG, initial TURBT results, CT results, and demographics [66,67]. The vesicle imaging reporting and data system (VI-RADS) scoring method further attempts to integrate MRI imaging to score MIBC and a recent analysis indicated a good sensitivity (0.83) and specificity (0.90) [68,69]. This possibility was previously found to allow discrimination between groups that could and could not optimally benefit from TURBT, as well as agree with interobserver readings in MIBC diagnosis [70,71]. However, these population-based scoring systems only attempt to predict recurrence after treatment based on previous, aggregated patterns and may not be as accurate in predicting individual response to therapy. To rectify this shortcoming, further involvement of neoadjuvant chemotherapy data (nacVI-RADS) has shown some promise in a small group of 10 patients at predicting response, indicating that integration of chemoradiotherapy statistics into existing scoring systems may provide prognostic power [72]. In spite of this progress, more precise benchmarks/biomarkers that do not require invasive repeat TURBT or frequent CT/MRI scanning are needed to measure the response to therapy in as close to real-time as possible. 

### 5.1. Current and Ideal Biomarkers

Typical molecular markers for bladder cancer are BTA, NMP22, and microsatellites (when examining tumor tissue for mismatch repair efficacy) [15]. For MIBC, recent forays into multi-genomic approaches using weighted-gene network analyses have indicated CLK4, DEDD2, ENO1, and STYL1 as genes of interest in addition to Lynch syndrome-associated and checkpoint-associated genes (FGFR3, PIK3CA, TERT, p53, MSH2, MSH6, MLH1, and PMS2) [3,5,8,73]. However, these studies primarily rely on solid tumor tissue analyses after biopsy to check for mutations caused by mismatch repair defects and represent only static snapshots of tumor status. New approaches that check peripheral blood for circulating tumor cells or exosomes, from which genetic analyses can be conducted, may offer more resolution into the progression of MIBC metastasis [74,75]. These approaches are possible with current technology and offer a low-cost, high-throughput lab method adaptable from centrifugal precipitation and microbead protocols used for central nervous system-derived exosomes [76]. Transcriptomic profiling of patients has attempted to overcome the heterogeneous nature of bladder cancer by broad classification into molecular classes [77]. Meanwhile, higher fidelity scanning methods using multi-omics approaches are attempting to categorize metabolic and molecular subtype profiles for MIBC patients to predict non-responders to cisplatin, anti-PD-L1 and other chemotherapies [78]. 

This section will attempt to detail the most desirable characteristics of biomarkers, particularly the recent discovery of exosomes as a useful clinical tool for cancer profiling and possible adjuvant therapy. 

### 5.2. Liquid Biopsies: The Convenience of Frequent Sampling

The primary characteristic of a useful biomarker is the ability to frequently sample, allowing the tracking of tumor progression with regard to both internal (i.e., microenvironment) and external (i.e., metastatic potential) statuses. This is important as larger amounts of data translate into better prognostic ability for not only individuals but entire populations as aggregate case data can be compared to large control datasets from nested populations (nested case controls) [79]. However, for bladder cancers, cystoscopic biopsy, CT/MRI, and other tissue sampling/scanning methods are time/resource intensive and may create pain and inconvenience for the patients. 

In light of the need for accurate, frequent sampling with minimal invasiveness and cost, the concept of liquid biopsies (sampling of circulating tumor cells [CTCs] or exosomes obtained from bodily fluids [e.g., blood, urine]) has seen increased development towards clinically precise separation and analysis methods [80]. Of note is the PredicineBEACON survey, which has already reported the utility of liquid biopsy methods (blood and urine) for individual mutation analysis profiling, demonstrating high precision, accuracy, and sensitivity in rapid evaluation of PIKC3A, FGFR3, and TERT mutations associated with MIBC during neoadjuvant therapy. This was in addition to mutational burden by copy number quantification (2022 ASCO. Abstract 539). 

Tumor-derived exosomes, which are nanoscale extracellular vesicles (EVs, 40–150 nm in size) released from cells as metabolic byproducts for communication or waste disposal, can be sampled from blood and are stable both in circulation and collection [81]. In particular, exosomes enriched in nucleic acids or proteins that indicate the genetic or metabolic state of tumors would provide valuable information over time that could reflect the impact of therapies on both tumor cell reproduction, metabolic status, and possible therapy resistance. 

The most crucial part of the exosome analysis process is collection and separation, as the blood/urine milieu contains large numbers of mixed solutes, cells, proteins, plasma, and immune components. Of note, several reports have detailed collection methods for urinary exosomes that could be exploited for bladder cancer detection and monitoring, including optimized ultrafiltration with 0.22 µm/10 kDa filters as well as ultracentrifugation, filtration, and protease treatments [82,83]. These low-cost, high-recovery methods use existing technology and are easily adapted for high-throughput clinical lab analysis. As for blood, exosome collection by ultrafiltration, magnetic bead/immunoaffinity capture, and ultracentrifugation have been used to profile diverse other cancers and can easily be applied to bladder cancer with the advantage of capturing CTCs in microcavity or other microfilter systems for culture expansion and profiling [84,85,86]. Combined with solid tumor samples from cystoscopic biopsy and predictive CT/MRI data, exosome/CTC sampling may be a low-cost and rich source of biomarkers to supplement and reinforce conclusions on prognosis and therapy response in bladder cancer patients. 

### 5.3. Amenable to Profiling

The ideal biomarker is also easy to profile with existing technology, especially -omics-based methods. Urinary exosomes, as representative of the bladder milieu, immune interactions, and cellular interactions, are a superior choice [87] (10.3390/pharmaceutics14102027). Starting from an initial 2010 report by Welton et al. in cultured cells, a recent study examined urinary exosomes from patients scheduled for RC and found mass spectrometry to be easily accomplished once a decoy search database and two-peptide matching filters were applied [88,89]. Beads with anti-CD63 were then used to capture free proteins, including CD9, CD63, Rab, heat shock proteins, and CD81 [89]. Thus, proteomic approaches incorporating mass spectrometry could provide high-throughout analysis for clinical applications.

Meanwhile, recent reports have isolated urinary exosomes and found clinically significant DNA amounts amenable to genomic profiling. Lee et al. found that urinary exosomes from 9 patients were similar to cell-free DNA in profiling of mutations, as well as higher correlation between these captured EVs and solid tumor status [90]. A subsequent and similar study by Zhou et al found in 2021 that, after treatment with DNAse I to remove extraluminal nucleic acids, urine and serum exosomes had similar particle trends, markers (flotillin-1 and CD9), and quality [91]. Of importance was their finding that Sanger sequencing was more sensitive than whole-exome sequencing for capturing mutation profiles and that mutation profiles and frequencies (featuring genes such as KLK10, PSCA, PTK2, ETV6, and TBX3) were reproducible and usually located in untranslated regions [91]. Although recent studies have been limited in patient numbers, the potential for commercial development of Sanger-based exosome profiling kits is high in light of this data. 

Transcriptomics-based approaches with exosomes are also possible. Discrimination between benign prostate hyperplasia and prostate cancer, for which prostate-specific antigen is insensitive, was achieved through collection and microarray transcriptomic profiling of urinary exosomes to arrive at CDH3 as a definitive biomarker [92]. The impact of RNA on diagnosis was also reinforced by Zheng et al., who detected exosome non-coding RNA for PTENP1 which was secreted by normal cells to target bladder cancer cells and increase apoptosis [93]. Along these lines, RNA analysis could provide valuable insight into transcriptional factors when combined with proteomic data. Thus, transcriptomics, in addition to proteomics and genomics, may serve as a mature platform for exosome and potential CTC analysis and profiling.

### 5.4. Provides Targeting and Therapeutic Options

Since exosomes are a natural cell-to-cell communication paradigm, they can be readily exploited for delivery of therapeutics. Natively, the endosomal sorting complex (ESCRT 0–3) generates exosomes that are uptaken by diverse mechanisms, including pinocytosis, plasma membrane fusion, or endocytosis [94]. The cargo capacity, although limited by size, can contain nucleic acids, proteins, biometabolites (e.g., sugars, vitamins), and lipids (e.g., cholines, steroids) [94]. It may also be possible to load photosensitive dyes for NIR therapy into these exosomes. Additionally, small chemical molecules, miRNA, or targeted peptides can be loaded and produced en masse for therapies and encased in an exosome package that specifically binds to tumor cells using exosomal surface proteins like LAMP-2B [95,96]. The higher metabolism of tumor cells and more acidic extracellular pH (which itself can be manipulated through drugs or proton pump inhibition) would help facilitate engineered exosome uptake by promoting fusion with the cell membrane [96,97].

Thus, exosomes, coupled with CTCs (if they can be captured efficiently), may provide a singular solution for generating rapid, accurate, and individualized metabolic and genetic profiles of tumors that allow for (1) prediction of therapy response, (2) changes in dosages to compensate for anticipated resistance, (3) screening for metastasis, and (4) delivery of customized drugs or miRNAs to silence key resistance genes, induce apoptosis, or reduce metastatic/growth potential of tumor cells (Figure 2). Although several RNA and natural compound clinical trials are ongoing, no current bladder cancer exosome drug delivery trial has been completed although 3 have been registered (ClinicalTrials.gov NCT04155359, NCT05270174, NCT05559177) [98]. Future studies in large populations with exosome-based screening, plus engineered exosomes to deliver drugs in conjunction with standard-of-care therapy for bladder cancer, will be instructive as to the full potential of these nanoparticles.

## 6. Conclusions

The past 20 years has seen a rise in the overall survival of MIBC due to the advent of multimodal therapy. In particular, combinations of chemotherapy, ICIs, radiation, and surgery (RC/TURBT) have increased both disease-free survival periods as well as overall survival up to 10 years. Additionally, new, immune-based targeting with antibodies and lectin are also being fast-tracked from the lab bench to the bedside. However, the lack of biomarkers to regularly examine the progress of tumors and track their metabolic, genomic, and metastatic conditions have hampered progress in achieving more beneficial outcomes. With the advent of exosome and CTC collection and analysis, frequent sampling for accurate profiling as well as engineered drug delivery are now possibilities to be explored. Successful TMT may thus evolve into quadrimodal therapy (QMT) as TURBT, chemotherapy, radiation, and ICI/ADC combinations are evaluated, profiled, and assisted by tumor exosome analysis and engineered exosome treatment (Figure 2).

**Contribution to the Field:** While bladder cancer diagnosis and treatment have undergone refinement and incremental improvement over the last 20 years, neo-adjuvant therapy remains underutilized and 10-year survival rates have not demonstrated dramatic gains. New therapies that preserve quality of life, such as bladder-sparing surgeries, are being investigated along with novel therapeutic modalities that could synergistically increase tumor control and prevent metastasis. However, the lack of effective biomarkers renders the real-time evaluation of tumor status impossible and treatment paradigms are reduced to reactive guesswork instead of accurate prognosis. Exosome sampling may provide some relief for the clinician as analysis of these tumor-produced, nanoscale particles can be easily collected, profiled, and used to customize therapy. Additionally, they can be engineered to deliver small molecular cargoes to tumor cells, opening up a new field of therapy. This paper contributes to the field by overviewing current treatments, analyzing new treatments, and exploring the new field of exosomes as both biomarker sources and treatment options.

## Figures and Tables

**Figure 1 biomedicines-11-00539-f001:**
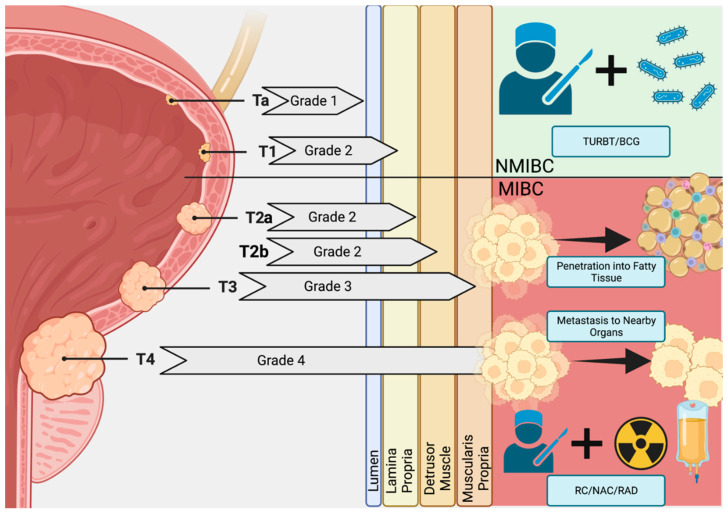
Bladder Cancer Pathogenesis. Early in situ carcinomas mature into Ta (Grade 1) and T1 (Grade 2) non-muscular invasive bladder cancers (NMIBC) malignancies that begin to penetrate the muscle wall of the bladder. Here, transurethral resection of the bladder (TURBT) and Bacillus Calmette–Guérin (BCG) therapy are often curative. However, a subpopulation is resistant to these therapies and the tumor penetrates the muscle wall and fatty tissue, becoming advanced-stage muscle-invasive bladder cancer (MIBC; Grades 3 and 4). With metastasis to surrounding organs, multimodal therapies featuring radical cystectomy (RC), neoadjuvant chemotherapy (NAC), and radiation (RAD) are used to increase survival by tumor growth and metastasis control. Created at BioRender.com.

**Figure 2 biomedicines-11-00539-f002:**
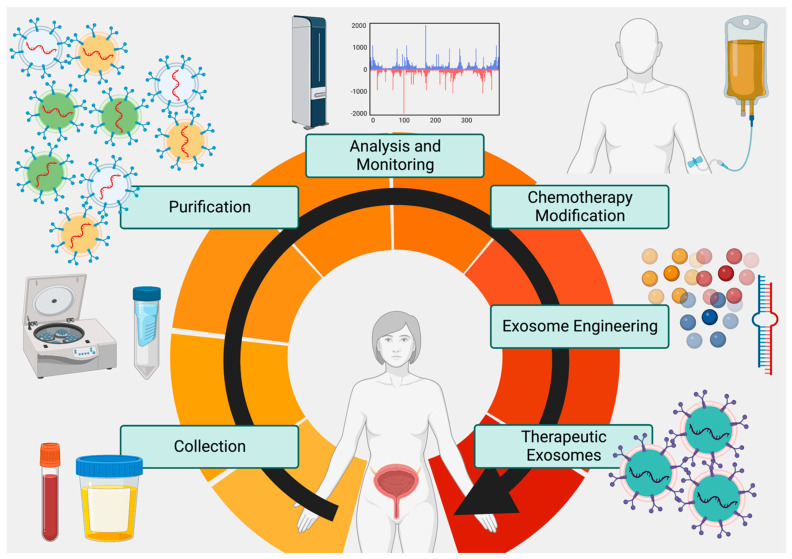
The Proposed Exosome Cycle. Tumor exosomes can be frequently collected from urine and blood, purified in high volumes, analyzed via -omics technology, and used to evaluate both tumor metabolism and response to therapy. In addition to customizing therapy based on exosome analysis, exosomes engineered to deliver custom payloads to tumor cells to promote chemoradiation susceptibility and apoptosis can be used as a synergistic adjunct to surgery and chemoradiation therapies. Created at BioRender.com.

**Table 1 biomedicines-11-00539-t001:** Treatment Outcomes for Trimodal Therapies (TMT) published after 2010.

Study	Phase and Follow-Up	Stage	N	Concomitant Chemotherapy	RT	OS	DFS	Salvage RC
Lagrange et al., 2011 [37]	II 8 yr	cT2-4a N0/Nx	51	Cisplatin + 5-FU×3	63 Gy ST	36% (8-yr)	-	33.3%
Choudhury et al., 2011 [38]	II 36 mo	cT2-3 N0/Nx	50	Gemcitabine weekly	52.5 Gy in 20	75% (3-yr) 65% (5-yr)	82% (3-yr) 78% (5-yr)	14%
James et al., 2012 [39]	III 69.9 mo	cT2-4a N0	182	5-FU, MMC×2	55 Gy or 64 Gy	48% (5-yr)	67% (2-yr)	11.4% (2-yr)
Tunio et al., 2012 [40]	III 5 yr	cT2-4 N0/Nx	200	Cisplatin weekly	65 Gy ST	52% (5-yr)	-	-
Zapatero et al., 2012 [41]	Retrospective 60 mo	cT2-4a N0	39	Cisplatin weekly (paclitaxel: *n* = 5)	64.8 Gy ST	73% (5-yr)	82% (5-yr)	33%
Giacalone, et al., 2017 [42]	Retrospective 7.21 yr (median)	cT2-4a N0M0	575	Cisplatin, 5-FU, gemcitabine, et al., varied in different protocols	44–66 Gy, varied in different protocols	57% (5-yr) 39% (10-yr)	84% (5-yr, 2005–2013) 60% (5-yr, 1986–1995)	16% (5-yr, 2005–2013) 42% (5-yr, 1986–1995)
Kulkarni, et al., 2017 [43]	Retrospective 4.51 yr (median)	T2-4 N0/Nx	56	4 cycles of Gemcitabine plus Cisplatin	66 Gy	6.61 yr (median)	76.6% (5-yr)	10.7% (5-yr)

OS = overall survival, DFS = disease-free survival, RT = radiotherapy, RC = radical cystectomy, 5-FU = 5-fluorouracil, MMC = mitomycin C.

## Data Availability

No new data were created or analyzed in this study. Data sharing is not applicable to this article.

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
