# Peer review of "The Current Progress and Future Options of Multiple Therapy and Potential Biomarkers for Muscle-Invasive Bladder Cancer"

_biomedicines, 2023, doi:10.3390/biomedicines11020539_

Round 1

Reviewer 1 Report

With this review authors aimed to overlook the present diagnostic, therapeutic and prognostic features available for the management of the muscle-invasive cancer bladder. They also tried to suggest new ways to predict the prognosis and the developement of the disease using new biomarkers. Authors analysed first the imaging techniques used for the diagnosis (i.e. Magnetic Resonance Imaging and Computer Tomography) and the conventional exams like cystoscopy, then they focused on treatment paradigms making a recapitulation on traditional therapies and analysing the newest treatments like the Immune Checkpoints Inhibitors (ICIs). The last part of this work is about the identification of a reliable biomarker to monitorize the effectivness of the previous therapy and the probability of an overlap. At this regard authors deepened the genetic analysis on exosomes and Circulating Tumor Cells (CTCs) finding them a reliable live photography of the disease situation.  

My suggestions:

English language should be corrected and improved, especially for the introduction of the work. The writing is not very clear in many parts.

Authors should add “MIBC diagnosis” and “MIBC treatment” in the keywords. The work doesn’t treat only exososomes and biomarkers

Given the huge incidence of MIBC in the population, the use of exosomes as a reliable biomarker for the developement of the disease sohuld be accessible to every patient suffering from this cancer but no mention is done in the work about the costs of the methodic or the required technologies to apply it. Please deepen this aspect

Analysing the lines 141-143 is said that “a recent trial is currently evaluating multiparametric MRI for replacement of TURBT staging”. MRI is getting more and more important in diagnosis and prognosis of MIBC, specially if matched to other parameters; for example at this regard I can suggest the analysis of this article: https://pubmed.ncbi.nlm.nih.gov/34258022/

The use of miRNAs as a therapeutic option is not universally accepted yet, there are few RCT to support this therapeutic strategy. Further RCTs are required 

Treatment with ICIs is not applicable to all patients because the response to treatment can be not satisfying. There are models to predict the response to ICIs and select the correct patients, at this regard i can suggest the analysis of this work: https://pubmed.ncbi.nlm.nih.gov/35626149/

Author Response

Response to Reviewer 1 Comments

Point 1: English language should be corrected and improved, especially for the introduction of the work. The writing is not very clear in many parts.

Response 1: We apologize for the language in the manuscript and have subjected the entire work to a complete editing by a native speaker familiar with the topic. A Certificate of Proofreading has also been submitted to the Editorial Office.

Point 2: T Authors should add “MIBC diagnosis” and “MIBC treatment” in the keywords. The work doesn’t treat only exososomes and biomarker

Response 2: Thanks for your suggestion. We searched the MeSH and added “diagnosis” and “therapeutics” in the keywords.

Point 3: Given the huge incidence of MIBC in the population, the use of exosomes as a reliable biomarker for the development of the disease should be accessible to every patient suffering from this cancer but no mention is done in the work about the costs of the methodic or the required technologies to apply it. Please deepen this aspect.

Response 3: Exosome isolation and analysis are possible with current technologies and recovery methods are based around centrifugation for isolation and then microbead adhesion to enrich exosomes for standard genetic analysis (10.3389/fnmol.2020.00038). We have added an explanatory sentence about this method in the text. Section 5.2 has also been modified to indicate the ease and low-cost nature of the method. Section 5.3 already indicates the ease of MS-MS analysis on urinary exosomes and that lower-cost Sanger sequencing is sufficient for analysis.

Point 4: Analyzing the lines 141-143 is said that “a recent trial is currently evaluating multiparametric MRI for replacement of TURBT staging”. MRI is getting more and more important in diagnosis and prognosis of MIBC, specially if matched to other parameters; for example, at this regard I can suggest the analysis of this article: https://pubmed.ncbi.nlm.nih.gov/34258022/

Response 4: Thanks for your suggestion. We agree that constant development of MRI is an important aspect in both diagnosis and management of MIBC with regard to treatment so we added this reference.

Point 5: The use of miRNAs as a therapeutic option is not universally accepted yet, there are few RCT to support this therapeutic strategy. Further RCTs are required.

Response 5: With the advent of engineered exosomes, miRNAs are gaining the possibility of becoming a valid therapeutic option. Although no definitive studies have been conducted on a large scale that exploit miRNA for MIBC therapy, we wanted to indicate ALL possibilities to stimulate development and interest in this area as we believe it to be part of tailored and customized medicine for better clinical outcomes.

Point 6: Treatment with ICIs is not applicable to all patients because the response to treatment can be not satisfying. There are models to predict the response to ICIs and select the correct patients, at this regard i can suggest the analysis of this work: https://pubmed.ncbi.nlm.nih.gov/35626149/

Response 6: Thanks for your suggestion. The response to ICI is definitely an important aspect in the overall survival rates encountered with MIBC and we thus added this reference.  We also tailored the paragraph around the concept of the predictive model as indicated by the reviewer.

Reviewer 2 Report

Dear Editor, 

This review provides an overview of the pathogenesis and progression of bladder cancer, standard treatments and outcomes, some novel treatment studies, and the current status of biomarker and therapy development.

In my opinion, the review contains too much information on pathogenesis and on current status than  novel treatment and markers

I believe that the author should better balance the review by reducing the current status and improving the novelty 

In particular

The novel treatment of checkpoint blockade is should be discussed before the future treatment option and novel markers 

Frequent Sampling is not the correct name for a paragraph  it is confounding if you discuss novel biomarkers: chapter should indicate clearly what is discussed

Moreover, in some paragraphs, such as  therapeutic and diagnostic approaches, the discussion is very dispersive because the authors collect all the problems that come from the use of exosomes as  content, source,  use as prognostic, diagnostic, and drug delivery without insufficient  information to understand the problems and if they are tested in the bladder as a marker

Author Response

Response to Reviewer 2 Comments

Point 1: In my opinion, the review contains too much information on pathogenesis and on current status than novel treatment and markers. I believe that the author should better balance the review by reducing the current status and improving the novelty

Response 1: We apologize for the length of the introduction but we wanted our review to completely outline the pathogenesis, especially genetic, and current conditions for readers who might not be experts in the field. In this way, we hoped to make our paper attractive to a broader readership. We have gone through and tightened up the text to better balance the paper. Unfortunately, a lack of novel treatments (other than combinations of existing therapies being tested in RCTs) makes the novelty rather sparse and we apologize for this shortcoming. However, collating and reporting what novelty does exist may stimulate more development of new therapies.

Point 2: The novel treatment of checkpoint blockade is should be discussed before the future treatment option and novel markers.

Response 2: Thanks for your advice. We have rearranged the order.

Point 3:Frequent Sampling is not the correct name for a paragraph. It is confounding if you discuss novel biomarkers: chapter should indicate clearly what is discussed.

Response 3: We apologize for this confusion. We have renamed this chapter to better indicate the focus on liquid biopsy as a method of tracking progression.

Point 4:Moreover, in some paragraphs, such as therapeutic and diagnostic approaches, the discussion is very dispersive because the authors collect all the problems that come from the use of exosomes as content, source, use as prognostic, diagnostic, and drug delivery without insufficient information to understand the problems and if they are tested in the bladder as a marker.

Response 4: We apologize for the confusion in these sections. The beginning of the paper was set up to showcase the current paradigms and the shortcomings of reactive imaging and evaluation in the face of a fast-changing malignancy such as MIBC. Exosomes and circulating tumor cells, in contrast, show high potential for rapid and accurate sampling of tumor status. This has been reported by multiple studies and an excellent review of the topic by Walker et al. in MDPI Pharmaceutics (10.3390/ pharmaceutics14102027) is definitive for the details. However, we did not want to duplicate too much information from this paper and will instead add it as a reference for the reader to pursue.

Reviewer 3 Report

Manuscript entitled "The Current Progress of Multiple Therapy and Potential Biomarkers for Muscle-Invasive Bladder Cancer"

1. This article doesn't include some updated targeted therapy (ex. FGFR targeted therapy, ERBB2 targeted therapy, and other protein-targeted therapy).

2. The molecular profiling characters of UC should be discussed in more detail.

Author Response

Response to Reviewer 3 Comments

Point 1: This article doesn't include some updated targeted therapy (ex. FGFR targeted therapy, ERBB2 targeted therapy, and other protein-targeted therapy).

Response 1:  Thanks for your advice. We have added a paragraph describing FGFR targeted therapy, which is commercially available now.

Point 2: The molecular profiling characters of UC should be discussed in more detail.

Response 2: Thanks for your advice. We tried to add more details in this part.

Round 2

Reviewer 1 Report

Authors answered all comments and suggestions.

Author Response

Thank you for your suggestion, we really appreciate your help!

Reviewer 3 Report

The revision is accepted in the present form.

Author Response

Thank you for your suggestion, we appreciate your help!